# Immunosuppressive Effects of Myeloid-Derived Suppressor Cells in Cancer and Immunotherapy

**DOI:** 10.3390/cells10051170

**Published:** 2021-05-11

**Authors:** Mithunah Krishnamoorthy, Lara Gerhardt, Saman Maleki Vareki

**Affiliations:** 1Cancer Research Laboratory Program, Lawson Health Research Institute, London, ON N6A5 W9, Canada; mithunah.krishnamoorthy@lhsc.on.ca (M.K.); lgerhard@uwo.ca (L.G.); 2Department of Pathology and Laboratory Medicine, University of Western Ontario, London, ON N6A 3K7, Canada; 3Department of Oncology, Division of Experimental Oncology, University of Western Ontario, London, ON N6A 3K7, Canada

**Keywords:** immunotherapy, MDSCs, T-cells, anti-PD-1, cancer, tumors

## Abstract

The primary function of myeloid cells is to protect the host from infections. However, during cancer progression or states of chronic inflammation, these cells develop into myeloid-derived suppressor cells (MDSCs) that play a prominent role in suppressing anti-tumor immunity. Overcoming the suppressive effects of MDSCs is a major hurdle in cancer immunotherapy. Therefore, understanding the mechanisms by which MDSCs promote tumor growth is essential for improving current immunotherapies and developing new ones. This review explores mechanisms by which MDSCs suppress T-cell immunity and how this impacts the efficacy of commonly used immunotherapies.

## 1. Introduction

Myeloid cells make up the largest group of nucleated cells of hematopoietic origin [1]. Myeloid cells are a critical part of the innate immune system in healthy individuals, functioning to clear infections and remodeling tissues. However, in cancer, myeloid cells are reprogrammed by tumor-derived factors to suppress anti-tumor immune cells and promote tumor growth [2]. These reprogrammed myeloid cells are present in the peripheral blood and tumors of cancer patients and can diminish the anti-tumor effects of immunotherapeutics [3]. Consequently, this subset of myeloid cells was coined as myeloid-derived suppressor cells (MDSCs) due to their myeloid lineage and capability to suppress immune responses [4]. During the course of tumor progression in many cancer types, MDSCs may increase by ten-fold in the peripheral blood and play a role in many non-immunological functions, such as angiogenesis and the formation of tumor metastases [5,6]. Furthermore, the stimulation and activation of these cells are a common feature of progressive cancers [7]. These are some of the mechanisms by which MDSCs promote tumor growth.

MDSCs are developmentally immature and are present in various stages of myelopoiesis. These cells are typically identified in mice by their co-expression of CD11b (integrin α M) and GR-1. Anti-GR-1 antibody treatment of immunocompetent mice with ultraviolet light-induced tumors reduced tumor growth, which was accompanied by the reduction of CD11b^+^ cells [8,9]. This observation indicates that the target cells expressed both GR-1 and CD11b, and they were protective of tumors. MDSCs in mice can be further categorized into two groups based on biological properties: polymorphonuclear/granulocytic (PMN)- and the monocytic (M)-MDSC subsets. In humans, there are three subcategories of MDSCs, namely PMN-MDSCs, M-MDSCs, and immature (i)-MDSCs, which lack the lineage markers of either PMN-MDSCs or M-MDSCs [10]. The murine equivalent of i-MDSCs has not been identified. More advanced techniques such as single cell RNA-seq and mass cytometry have further strengthened our understanding of MDSC subsets and has highlighted the heterogenic nature of the MDSC population [11,12,13]. For instance, single cell transcriptomics of splenocytes from mouse models of breast cancer reveal the existence of a subset of cells that express genes found in both M-MDSCs and PMN-MDSCs [12]. The identification of one or more specific subsets of MDSCs can also lead to the identification of biomarkers that can predict poor prognosis. In glioblastoma patients, mass cytometry by time of flight (CyTOF) analysis of peripheral blood identified that an immune cell signature consisting of high frequencies of CD33^lo^ MDSCs, and low frequencies of CD8^+^ T-cells and DCs positively correlated with poor prognosis [14].

PMN-MDSCs share many morphological characteristics to neutrophils and makeup 80% of MDSCs in all cancer types [2]. The abundance of these cells in cancer is often correlated with resistance to various therapies, poor prognosis, and shorter overall survival in patients [15]. In mice and humans, PMN-MDSCs are defined by the surface expression of CD11b^+^Ly6G^+^Ly6C^low^ and CD14^−^CD11b^+^CD15^+^(CD66b^+^), respectively [16]. These cells mainly promote tumor progression by inducing tolerance in antigen-specific T-cells. PMN-MDSCs can produce reactive oxygen species (ROS) that eventually leads to the alteration of T-cell receptors rendering T-cells less responsive to their cognate antigen [17].

While significantly less studied, M-MDSCs also contribute to tumor progression. Similar to PMN-MDSCs, M-MDSCs are identified differently in both mice and humans. M-MDSCs express CD11b^+^Ly6G^−^Ly6C^hi^ in mice and CD14^+^HLA-DR^−/lo^ in humans [18]. In contrast to PMN-MDSCs, M-MDSCs produce nitric oxide (NO) to suppress immune activation [19]. High levels of M-MDSCs correlate with higher circulating tumor cells and metastasis in breast cancer [20]. Increased M-MDSCs were also significantly correlated with estrogen receptor-negative disease and worse overall survival, indicating that M-MDSCs may be a potential biomarker of more aggressive breast cancer [20]. Additionally, M-MDSCs can differentiate into immunosuppressive macrophages [21]. These M-MDSC-derived macrophages suppressed T-cell activation in vitro, similar to tumor-associated macrophages (TAMs), and they were dissimilar to macrophages differentiated from monocytes [21].

The exact mechanism that leads to the generation of MDSCs is still not completely understood. While it is known that MDSCs are derived from hematopoietic stem cells, the signals that prevent the normal development of myeloid cells and promote the development of suppressive cells are still debated [22]. The delineation of this mechanism is important, as it can provide insight into how to prevent the generation of MDSCs and subsequently improve anti-tumor immunity. The presence of several cytokines and growth factors in the tumor microenvironment (TME), which are released by immune cells and tumor cells, promotes the development of MDSCs. Those soluble factors include granulocyte–macrophage colony-stimulating factor (GM-CSF), granulocyte CSF, macrophage CSF, stem cell factor, transforming growth factor (TGF)-β, tumor necrosis factor (TNF)-α, vascular endothelial growth factor (VEGF), prostaglandin E2 (PGE2), and cyclooxygenase 2 [23]. Many of these factors trigger the activation of Janus kinase (JAK) protein family members and signal transducer and activator of transcription 3 (STAT3), which are signaling molecules involved in cell survival, proliferation, differentiation, and apoptosis [24]. Evidence of this is observed in tumor-bearing mice that have increased levels of phosphorylated STAT3 in MDSCs compared to immature myeloid cells from naive mice. As such, specific inhibitors of the JAK/STAT pathway were found to eliminate MDSCs in mice, leading to improved differentiation of dendritic cells and ultimately a decrease in tumor burden [25]. MDSCs present in patients with pancreatic ductal carcinoma were found to be regulated by STAT3, and those with higher frequencies of STAT3^+^ MDSCs were correlated with shorter overall survival. In vitro, STAT3^+^ MDSCs demonstrated potent immunosuppressive effects on T-cell activation compared to MDSCs treated with a STAT3 inhibitor [26]. This suggests that STAT3 regulates an immunosuppressive phenotype in MDSCs and is a potential biomarker of poor prognosis in patients with advanced disease.

The JAK/STAT signaling pathway is upstream of the regulator of CCAAT/enhancer-binding protein beta (C/EBPβ), which is a transcription factor that is heavily implicated in the differentiation of MDSCs. Upregulation of C/EBPβ is necessary for emergency granulopoiesis and the de novo generation of myeloid-derived neutrophils during systemic infection [27]. STAT3 activates its downstream target C/EBPβ in response to GM-CSF and contributes to the expansion of CD11b^+^Gr1^+^ cells in mice [28]. Importantly, C/EBPβ deficiency affects the differentiation of MDSCs, mostly M-MDSCs. The adoptive transfer of antigen-specific CD8^+^ T cells lead to the regression of established fibrosarcomas in mice lacking C/EBPβ. This suggests that C/EBPβ is important in regulating immunosuppression and could be a target for cancer therapy [29].

The development of either M-MDSCs or PMN-MDSCs is also regulated by different apoptotic pathways. More specifically, the anti-apoptotic molecule Cellular FLICE (FADD-like IL-1β-converting enzyme)-inhibitory protein (c-FLIP) was found to inhibit necroptosis and Caspase-8-mediated apoptosis and is necessary for the development of M-MDSCs [30,31]. Interestingly, the overexpression of c-FLIP in M-MDSCs leads to increased immunosuppressive capabilities [32]. PMN-MDSC development is regulated by another anti-apoptotic molecule, myeloid leukemia cell differentiation protein (MCL-1). The depletion of MCL-1 in mice leads to an absence of PMN-MDSCs [30].

## 2. T-Cell Suppression by MDSCs

All MDSCs are potent suppressors of anti-tumor immunity, mainly by inhibiting T-cell functions. MDSCs within the tumor can directly regulate T-cell functions by engaging their inhibitory receptors (Figure 1). MDSCs within the TME express high levels of Fas-ligand (FasL), capable of inducing apoptosis of tumor-infiltrating lymphocytes (TILs) [33]. PMN-MDSCs specifically induce CD8^+^ T-cell apoptosis through the FasL–Fas axis, which results in local T-cell suppression [34]. Moreover, MDSCs can induce T-cell suppression by engaging negative checkpoint regulators programmed cell death protein 1 (PD-1), cytotoxic T-lymphocyte-associated protein 4 (CTLA-4), and T-cell immunoglobulin and mucin domain-containing protein 3 (TIM3). Importantly, MDSCs express the PD-1 ligand (PDL1) and Galectin-9, which bind PD-1 and TIM3 on TILs, respectively, restraining their anti-tumor immune response [35,36,37].

These MDSCs also produce high levels of IL-10 and TGF-β, largely enhancing immune tolerance. In various tumor types, IL-10 has been shown to impair CD8^+^ T-cell function [38]. IL-10 production also correlates with the reduction of IL-12, which is a cytokine secreted by dendritic cells mainly involved in stimulating anti-tumor immunity [39]. Additionally, IL-10 produced by MDSCs has been implicated in the development of T regulatory cells (Tregs) [40,41]. Nearly 40 years ago, TGF-β was identified for its ability to impair effector T-cell proliferation by inhibiting IL-2 production [42,43]. TGF-β signaling can directly suppress CD8^+^ T-cell function by inhibiting the expression of genes encoding perforin, granzyme B, and interferon (IFN)-γ, which are essential to CD8^+^ T-cell cytotoxicity [44]. Interestingly, IL-10 and TGF-β, among others, can be packaged in exosomes by MDSCs, which are found to hyper-activate or exhaust CD8^+^ T-cells [33,45].

The expansion and activation of MDSCss occur under pathological conditions, such as hypoxia within the TME [46,47]. Under hypoxic conditions, hypoxia inducible factor 1α (HIF-1α) plays a crucial role in the induction of two ectonucleotidases, CD39 and CD73 [48]. MDSCs in the TME express high levels of these two ectonucleotidases, which, in tandem, convert ATP to adenosine, which is an immunosuppressive molecule [49,50]. Therefore, MDSCs can exploit the adenosinergic pathway to suppress anti-tumor activity, as adenosine functionally inhibits CD4^+^ and CD8^+^ T-cells, and stimulates regulatory immune cells [50].

A multi-level mechanism of negative T-cell regulation occurs through enhanced L-arginine metabolism. HIF-1α upregulation in response to hypoxia can also regulate MDSC differentiation and function, in part, by stimulating inducible nitric oxide synthase (iNOS) and Arginase 1 (Arg1) hyperproduction [47]. Both of these enzymes use L-arginine as a substrate, causing L-arginine depletion in the TME [51,52]. This leads to T-cell cycle arrest through silencing of cyclin D3, thereby disabling progression into the G1 phase [51]. L-arginine deficiency also appears to result in the loss of CD3ζ expression by preventing CD3ζ translation, resulting in T-cell dysfunction [53,54]. In addition to L-arginine depletion, Arg1-mediated local cysteine sequestration, metabolite production, and downstream signaling events further exert inhibitory effects on T-cells [1,55].

MDSCs highly express indoleamine 2,3-dioxygenase 1 (IDO1), causing intratumoral depletion of L-tryptophan, kynurenine production, and subsequently, the generation of lymphotoxic kynurenine metabolites [56,57]. IDO1 induces effector T-cell tolerance through cell-intrinsic and extrinsic mechanisms. Tryptophan deficiency stimulates general control nonderepressible 2 (GCN2), initiating integrated stress response, thereby inducing T-cell cellcycle arrest and anergy [58,59]. Moreover, GCN2 activation in this context mediates CD3ζ downregulation in CD8^+^ T-cells, which was associated with reduced effector function ex vivo [60]. Additionally, tryptophan depletion inhibits the activation of the mammalian target of rapamycin (mTOR) and protein kinase C (PKC), which triggers effector T-cell autophagy [61]. While both effector T-cells and Tregs require T-cell receptor (TCR) stimulation to exert their functions, the downstream activation of PKC has different effects on each population. The immunosuppressive capacity of Tregs appears to be negatively regulated by PKC recruitment and activation. Therefore, IDO1-mediated PKC inhibition enhances Treg function [62]. IDO1 activity in mice also promotes Tregs expansion through the direct conversion of CD4^+^CD25^−^ T-cells into CD4^+^CD25^+^ Treg cells, further mediating T-cell tolerance [63].

The activation of the kynurenine pathway is particularly harmful to effector T-cells, contributing to cancer immune evasion. The interaction between kynurenine and the aryl hydrocarbon receptor (AhR) of dendritic cells and macrophages stimulates their differentiation to a regulatory phenotype. In turn, these cells can induce the transdifferentiation of inflammatory T-helper-17 (Th17) cells to Tregs [64,65,66]. Similarly, this interaction can promote the differentiation of naïve CD4^+^ T-cells into Tregs through the induction of FoxP3 expression [64,65]. Together, AhR stimulation by kynurenine leads to the selective expansion of Tregs. Additionally, downstream signaling events of kynurenine-induced activation of AhR on CD8^+^ T-cells upregulate PD-1 expression on these cells [67]. Since tumor cells and MDSCs exploit the PD-1/PD-L1 checkpoint pathway, kynurenine mediates CD8^+^ T-cell anergy within the TME.

The consequences of the kynurenine pathways downstream of kynurenine production depend on a series of enzymatic reactions, such as the generation of active metabolites such as kynurenic 3-hydroxyanthranilic acid (3-HAA) and quinolinic acid. In addition to kynurenine, kynurenic acid is an endogenous ligand of AhR, further promoting Tregs expansion [68]. Kynurenic acid also inhibits CD4^+^ T-cell proliferation and IFN-γ production [69]. High concentrations of 3-HAA and quinolinic acid can inhibit the cell cycle progression of CD8^+^ T-cells and induce apoptosis in T-helper-1 (Th1) T-cells in a Fas-independent manner [70,71]. Additionally, quinolinic acid has pro-oxidative characteristics, as it can enhance ROS formation in the TME by forming complexes with iron (II) [72]. ROS is a well-recognized DNA-damaging agent that can impair T-cell proliferation and anti-tumor functions [73].

MDSCs facilitate the generation of free radicals through multiple mechanisms. MDSCs produce calcium-binding proteins S100A8 and S100A9, contributing to increased ROS production through binding and potentiating NADPH oxidase, specifically generating superoxide anions [74]. In the presence of water, hydrogen peroxide is produced, which is a potent suppressor of T-cell activity [75,76]. The S100 proteins can also stimulate MDSC induction and accumulation, acting as a positive feedback loop [39]. In addition to ROS, MDSCs mediate the production of the reactive nitrogen specie (RNS) peroxynitrite (PNT) through the reaction of NO and O_2_^−^ [77,78,79]. Molon et al. have shown that RNS induces the nitration of the chemokine CCL2, hindering T-cell infiltration into tumors. This restrained T-cell recruitment into tumors, leaving tumor-specific T-cells constricted to the stroma surrounding the tumor [80].

Tumor-derived factors within the TME induce cyclooxygenase-2 (COX-2) activity in MDSCs, which has been implicated in promoting IDO1, IL-10, ARG1, iNOS, and PGE2 [81]. PGE2 is a proinflammatory molecule with both positive and negative functions in terms of anti-tumor immunity. PGE2 promotes the final maturation of dendritic cells, increasing their stimulatory functions [82]. However, when present during dendritic cell development, PGE2 redirects precursor myeloid cells to develop an MDSC phenotype [83,84]. As such, there is a reduced frequency of dendritic cells capable of inducing effector T-cells and the accumulation of MDSCs, as well as Tregs can occur [83]. Studies have demonstrated that PGE2 inhibits T-cell proliferation through both IL-2-dependent and independent mechanisms [85,86]. Moreover, PGE2 suppresses the production of type-1 cytokines such as IFN-γ in both CD4^+^ and CD8^+^ T-cells, while enhancing the production of type-2 cytokines such as IL-4 and IL-10 [87,88,89]. PGE2 has also been shown to directly suppress CD8^+^ T-cell activation through the engagement of EP2 receptors [81].

Emerging evidence indicates that MDSCs contribute to the epithelial-to-mesenchymal transition (EMT). EMT is a biological process that reverts polarized epithelial cells to multipotent mesenchymal cells that have enhanced migratory capabilities and resistance to apoptosis [90]. Many of the immunosuppressive factors produced by MDSCs that contribute to effector T-cell suppression such as TGF-β, VEGF, IL-10, and IL-6 are shown to induce a stem-like phenotype in various tumor cells [91,92]. This enhances tumor cell plasticity, promoting survival, motility, and metastasis. Additionally, the EMT of tumor cells has been also been shown to correlate with immunotherapy resistance [93]. Thus, MDSCs and stem-like tumor cells are capable of augmenting their individual immunosuppressive effects, such as promoting TGF-β signalling, directly inhibiting the cytotoxic function of CD8+ T-cells, and polarizing various immune cells toward a regulatory phenotype [94,95,96].

As previously mentioned, MDSC subpopulations appear to exert their immunosuppressive functions through different mechanisms. M-MDSCs produce large quantities of NO and immunosuppressive cytokines, and demonstrate high Arg1 activity [97]. MDSC products and metabolites such as NO, Arg1, and cytokines are well known for their paracrine effects; therefore, M-MDSCs require cellular proximity. Conversely, PMN-MDSCs have been shown to require cell contact to exert specific immunosuppressive effects. These cells produce large amounts of ROS such as O_2_, H_2_O_2_, and PNT, which inhibit the function of tumor-specific CD8^+^ T cells [17,98]. Nagaraj et al. demonstrated that the production of PNT by MDSCs, when engaged with T-cells, caused the direct nitration of tyrosine residues within the TCR-CD8 complex [99]. They further showed that this impeded the binding of specific peptide MHC-I molecules to T-cells, rendering antigen-specific CD8^+^ T-cells unresponsive to antigen-specific stimuli as shown by reduced granzyme B and Ki67 in antigen-specific CD8^+^ T-cells [99]. Further evidence showed that tumor unspecific TCR complexes were not particularly affected, leaving those T-cells functional. Together, this indicates that PMN-MDSCs are capable of inducing antigen-specific CD8^+^ T-cell tolerance.

## 3. Role of MDSCs in Resistance to Immunotherapeutics

In the past decade, immunotherapies including immune checkpoint inhibitors (ICIs) and chimeric antigen receptor T-cells (CAR T-cells), have revolutionized cancer treatment. ICIs such as anti-PD-1 and anti-cytotoxic T-lymphocyte-associated protein 4 (anti-CTLA4) drugs function by reinvigorating chronically stimulated T-cells in the TME that demonstrate an exhausted phenotype. While treatment with ICIs, alone or in combination, can confer long-term survival in a significant fraction of cancer patients, many patients experience primary or acquired resistance to these treatments [100,101,102]. MDSCs contribute to immunotherapy resistance. In preclinical models of melanoma, MDSCs have been shown to reduce the efficacy of anti-PD-1 and anti-CTLA4 therapies [34]. As previously discussed, MDSCs induce T-cell apoptosis via Fas–FasL interaction. Zhu et al. demonstrated that FasL neutralization in combination with anti-CTLA4 and anti-PD-1 reduced tumor growth and T-cell apoptosis in tumor-bearing mice compared to those treated with immunotherapy alone [34]. They also analyzed the effects of PMN–MDSC depletion and FasL neutralization in the context of adoptive cell transfer with CD8^+^ T-cells. Data showed that FasL neutralization improved CD8^+^ T-cell infiltration and prevented TIL apoptosis mediated by PMN-MDSCs, thus increasing adoptive cell transfer efficacy in this model [34].

MDSCs may be used as a prognostic marker for the efficacy of immunotherapy with ICIs. Meyer et al. show that MDSC frequencies in the peripheral blood of metastatic melanoma patients correlate with their response to ipilimumab, an anti-CTLA-4 antibody [103]. Analysis of peripheral blood mononuclear cells (PBMCs) via flow cytometry, before and during treatment with ipilimumab, indicated that patients with low frequencies of circulating Lin^−^CD14^+^HLA-DR^-^ M-MDSCs showed an improved clinical response to this therapy [103]. These data confirm similar findings in other studies in metastatic melanoma patients treated with ipilimumab [104,105,106]. Current research is focused on identifying a reliable and clinically relevant method to use MDSC frequency as a biomarker for the clinical response to ipilimumab in melanoma patients [107]. In patients with advanced non-small-cell lung cancer (NSCLC), those who had progressive disease had higher percentages of PMN-MDSCs at baseline than those who showed a clinical response when treated with nivolumab, an anti-PD-1 antibody [108]. These clinical observations suggest that MDSCs contribute to primary resistance to immunotherapy regardless of cancer type or ICI administered.

The specific mechanisms of MDSC-mediated immunotherapy resistance in the previous studies were not explicitly stated; however, some studies have elucidated resistance mechanisms in their models. Gebhardt et al. demonstrated that in addition to elevated frequencies of MDSCs in the peripheral blood, MDSC activity was associated with poor clinical response in melanoma patients treated with ipilimumab. In this retrospective immune-monitoring study, a significant increase of M-MDSCs in the peripheral blood of non-responders was observed, while M-MDSC frequencies declined in responding patients compared to baseline values. These trends could be observed after the first ipilimumab treatment in this cohort. In assessing the production of NO by MDSCs and concentration of S100A8/A9 proteins in serum, non-responders displayed elevated levels of both molecules after the first infusion of ipilimumab compared with responders [109]. As NO and S100A8/A9 are employed by MDSCs and actively suppress anti-tumor immunity, this may indicate a mechanism by which MDSCs confer resistance to ICIs.

As TGF-β signaling is capable of amplifying the immunosuppressive processes within the TME and beyond, these pathways also display a putative role in ICI resistance [110,111,112,113]. In a study in which a subset of patients with urothelial cancer unresponsive to the ICI, anti-PD-L1, RNA sequencing revealed that TGF-β is associated with poor response [113]. Specifically, these patients expressed a TGF-β-induced cancer-associated fibroblast gene signature that was associated with an immune-excluded tumor phenotype. Using colon adenocarcinoma (MC38) and mammary (EMT6) mouse models, this study revealed that therapeutic blockade of TGF-β with antibodies promoted CD8+ T-cell inflammation and anti-tumor immunity, sensitizing tumors to PD-L1 therapy [113]. Evidence from another study using the MC38 mouse model revealed that anti-PD-1-resistant tumors exhibited reduced infiltration of effector T-cells and NK cells when treated with anti-PD-1. Mice displaying resistance were found to have active TGF-β and Notch signaling. Inhibiting both of these pathways during treatment with anti-PD-1 decelerated tumor growth in resistant tumors [114].

Specific evidence of MDSC-mediated immunotherapy resistance through TGF-β production has been demonstrated using a 4T1 mammary mouse model [114,115]. TGF-β neutralization was shown to promote anti-tumor activity of T-cells co-cultured with MDSCs. Moreover, the depletion of MDSCs diminished anti-tumor effects mediated by TGF-β neutralization [115]. These data indicate that TGF-β plays a vital role in the immunosuppressive activity of MDSCs and may contribute to the poor response associated with ICIs. Evidence using preclinical models suggests that combining specific TGF-β inhibitors with ICIs can facilitate effector T-cell infiltration and reduce the immunosuppressive myeloid compartment [113,116]. In turn, this stimulates anti-tumor immunity and mitigates ICI resistance.

MDSCs have been shown to contribute to the immunosuppressive nature of the TME, impairing the infiltration and anti-tumor function of endogenous as well as adoptively transferred and engineered immune cells. Chimeric antigen receptor (CAR) T-cell therapy is a form of personalized medicine that has shown promise in the treatment of hematological malignancies [117,118]. T-cells derived from a patient are genetically modified to express a membrane-bound protein that uses a single-chain variable fragment to recognize tumor-specific antigens in an MHC-independent manner. Upon this interaction, the CARs intracellular signaling domain can stimulate the T-cell, enabling the specific destruction of tumor cells [119].

MDSCs in patients with solid tumors such as sarcomas, glioblastoma, and non-squamous cell lung cancers, can promote resistance to CAR T-cell therapy. This can be exemplified by a study evaluating the efficacy of a third-generation anti-GD2-CAR T-cell therapy in preclinical models of neuroblastoma and pediatric sarcomas [120]. As GD2 is a surface antigen expressed by these tumors, anti-GD2-CAR T-cells with CD28 and OX40 co-stimulatory domains were assessed for their ability to mediate anti-tumor activity. Illustrating the importance of in vivo models, Long et al. found that these CAR T-cells effectively killed GD2^+^ neuroblastoma and osteosarcoma cells in vitro, but minimally showed any anti-tumor activity against osteosarcoma xenografts. Further studies revealed that the limited anti-tumor activity in this model was associated with enhanced frequencies of MDSCs in the blood, spleen, and the tumor [120]. However, the mechanism underlying this relationship has yet to be determined.

## 4. MDSCs and Microbiome

The microbiome has recently been identified as a potent modulator of anti-tumor immunity [121,122]. The microbiome is a collection of the genetic material of numerous bacteria, fungi, and other microbes with their surrounding environment [123]. While the human body harbors various microbial ecosystems, the most heavily studied is the gut microbiome due to the relative ease of accessing and analyzing gut bacteria from fecal samples. Gut bacteria have the potential to increase populations of immune cells while decreasing others. For example, the inoculation of mice with *Akkermansia muciniphila* was found to increase the frequency of CD4^+^ central memory T-cells in the draining lymph nodes and was associated with improved response to anti-PD-1 in mice bearing murine fibrosarcoma [121]. Melanoma patients with favorable responses to anti-PD-1 had an abundance of certain bacterial populations, namely *A. Muciniphila* and the Ruminococcaceae family members in their gut [121,122]. Increased abundances of these bacteria were correlated with decreased MDSC frequencies in peripheral blood prior to treatment. Furthermore, non-responders had increased abundances of bacteria in the order Bacteroidales, which was positively correlated with higher frequencies of MDSCs in the peripheral blood [122]. This observation suggests that MDSCs may be reduced by altering the gut microbiome.

The gut microbiome can be modified through fecal microbiota transplant (FMT), whereby the feces of one individual is administered either orally via capsules or rectally via enema to another individual. FMT is routinely used for the treatment of antibiotic-resistant *Clostridioides difficile* (C.diff) infections [124]. The efficacy of FMTs in reducing MDSCs was shown in mice harboring pancreatic tumors. These mice received FMTs from short-term survivors of pancreatic adenocarcinoma (PDAC), resulting in increased frequencies of MDSCs in the TME compared to mice that received FMTs from either long-term survivors of PDAC or healthy volunteers [125]. Consequently, mice receiving FMTs from short-term survivors had an increased tumor burden. In a phase I clinical trial, patients with PD-1-refractory melanoma were given FMTs from melanoma patients who had previously responded to anti-PD-1 therapy. Then, recipients of FMT were treated again with anti-PD-1 to observe whether resistance could be overcome by changing the gut microbiome. Interestingly, patients who overcame resistance after receiving FMT had a lower frequency of IL-8-expressing myeloid cells compared to patients who did not [126]. This shows that the composition of the microbiota can determine the presence of MDSCs, which can dictate tumor burden and affect survival.

Chemical manipulation of the gut microbiome also influences the recruitment of MDSCs. In a mouse model for cholangiocarcinoma, a malignancy of the bile duct, MDSCs show increased levels in the liver, gut, spleen, and lymph nodes compared to healthy mice. However, upon treating diseased mice with various broad-spectrum antibiotics, the accumulation of MDSCs decreased in the liver. The recruitment of MDSCs to the liver was correlated to Gram-negative gut commensal bacteria, which is increased due to increased gut permeability in patients with chronic gastrointestinal disorders. Hepatocytes secreted CXCL1, a chemokine responsible for PMN–MDSC recruitment, upon interaction with Gram-negative bacteria via their TLR4. Intrahepatic cholangiocarcinoma patient data also support the role of bacteria in tumor progression, which shows that those with lower expression of TLR4 have longer overall survival [127]. Gut dysbiosis that occurs after exposure to the carcinogenic compound 2,3,7,8-tetrachlorodibenzo-p-dioxin (TCDD) also leads to an accumulation of MDSCs in the peritoneal cavity of mice [128].

Disturbances in the initial development of the gut microbiome can play a role in the recruitment of MDSCs. Mice that were raised germ-free until weaning and then reconstituted with a complete microbiota were more susceptible to developing colitis-associated cancer in comparison to specific pathogen-free mice. Interestingly, the lack of a microbiota early in life leads to increased accumulation of PMN–MDSCs but not M-MDSCs in the colon. This was attributed to an increased expression of CXCL1 and CXCL2 in tumor tissues that could recruit PMN–MDSCs. Notably, administering a CXCR2 (the receptor for CXCL1 and CXCL2) blocking antibody resulted in reduced accumulation of PMN–MDSCs in the colon and subsequently prevented tumor development in inducible models of colitis-associated cancer [129].

Not only does the gut microbiome influence MDSC recruitment and function in the gastrointestinal tract but also malignancies of more distal organs. The microbiome produces metabolites that can disseminate throughout the body and influence immune cells at various sites. For example, the fermentation of dietary fiber by gut microbiota leads to the production of short-chain fatty acids (SCFAs), which can inhibit the destructive effects of ROS produced by MDSCs (Figure 2) [130,131]. SCFA-producing bacteria include members of the Bacteroidetes (Gram-negative) and Firmicutes (Gram-positive) phyla, which are the most abundant phyla in the human intestine [132]. In some cancers, such as glioblastoma, up to 5.4% of the tumor is made of MDSCs [133]. As SCFAs can readily cross the blood–brain barrier, the change in microbiota composition may mitigate the effect of MDSCs in brain tumors and could improve anti-tumor immunity [131]. Ultimately, the composition of the microbiome can heavily influence the recruitment and function of MDSCs and subsequently anti-tumor immunity.

## 5. Immunotherapies and Combination Strategies Targeting MDSCs

Given the immunosuppressive effects of MDSCs and their role in reducing the efficacy of existing immunotherapies, potential cancer treatment strategies might include (**a**) eliminating MDSCs from the TME, (**b**) preventing MDSC accumulation in the TME, or (**c**) mitigating their immunosuppressive functions. MDSCs are short-lived cells that are constantly being replaced by new MDSCs; therefore, depletion strategies need to keep up with renewal rates of these cells [134]. However, some approaches have shown promise (Figure 2).

Some chemotherapeutic agents have leukodepletion properties [135]. Gemcitabine, 5-fluorouracil (5-FU), and pemetrexed are among chemotherapeutics that can reduce the abundance of MDSCs in the tumor and peripheral organs of patients with various cancers [136,137,138]. This effect is augmented when combined with other therapies such as adoptive cell transfer of cytokine-induced killer cells in patients with pancreatic cancer and renal cell carcinoma, or VEGF inhibitor, bevacizumab in patients with non-small cell lung cancer [136,137,138]. It is worth acknowledging that some chemotherapeutics, such as CPT-11, have been associated with increased frequencies of MDSC populations. Kanterman et al. demonstrated that CPT-11 treatment caused MDSC accumulation and enhanced immunosuppression compared to 5-FU, by desensitizing MDSCs to apoptosis and maturation [137]. Together, these results indicate that different chemotherapies can preferentially affect MDSCs, but specific agents may be used to reduce their frequency.

The anti-GR-1 antibody was successful in depleting MDSCs in mice and subsequently reducing tumor burden [8]. A more targeted approach in depleting MDSCs is to use peptibodies, a protein consisting of the Fc portion of an IgG2b antibody fused to an MDSC-binding peptide. One study found that peptibodies using the S100 protein successfully depleted both PMN–MDSCs and M–MDSCs from mice harboring various tumors and inhibited tumor growth, unlike anti-GR1, which only depleted PMN–MDSCs. The peptibodies can opsonize the MDSCs and mark them for elimination via antibody-dependant cell phagocytosis. The use of these peptibodies did not affect other myeloid cells, such as dendritic cells or immature myeloid cells [139].

MDSCs along with Tregs can also be eliminated with the administration of IL-2 and anti-CD40, which initiates apoptosis of these cells in a Fas-dependent manner [140]. CD40 is a cell surface molecule that regulates the activation of antigen-presenting cells by binding to its natural ligand‑CD40‑on T-cells. Anti-CD40L can agonistically bind, circumventing the need for T-cells to activate antigen-presenting cells [141]. MDSCs also express elevated levels of TNF-related apoptosis-induced ligand receptors (TRAIL-Rs), which can be leveraged for potential therapies. When TRAIL-Rs were targeted with an antagonist antibody, tumor growth was delayed when used in both in vitro and in vivo settings [142]. However, as MDSCs in mice differ from human MDSCs in characterization, correlating murine experiments to human studies is challenging.

Human MDSCs were successfully targeted with gemtuzumab ozogamicin (GO), a humanized antibody against CD33 conjugated to a cytotoxic agent. When cultured with CD33^+^ MDSCs, GO was rapidly internalized by this population. While gemtuzumab on its own did not affect MDSCs survival, GO exhibited a dose-dependent effect on cell viability. Electron microscopy revealed bleb-like structures on the surface of treated MDSCs, confirming their death via apoptosis. In co-culturing T-cells with untreated or GO-treated MDSCs from various malignancies, T-cells co-cultured with GO-treated MDSCs exhibited greater proliferation than those co-cultured with untreated MDSCs. This shows that GO can indirectly increase T-cell activation and can potentially be combined with CAR T-cell therapy. Indeed, GO-mediated depletion of MDSCs enhances CAR T-cell killing of target glioblastoma and mesothelioma tumor cells in vitro [143]. The consequences of MDSCs reducing CAR T-cell efficacy are currently being investigated. Accordingly, many methods that hinder MDSCs, including depletion, functional inhibition, and migration blockade, are being evaluated in combination with CAR T-cell therapy.

Recently, Sun et al. demonstrated that blocking MDSC recruitment enhances the efficacy of CAR T-cell therapy in a mouse model of breast cancer [144]. Olaparib, a poly (ADP-ribose) polymerase inhibitor (PARPi), was shown to suppress MDSC migration, increasing the anti-tumor activity of epidermal growth factor receptor (EGFR) *v*III-targeted CAR (806-28Z CAR) T-cells. Olaparib reduced the expression of chemokine CXCL12α by cancer-associated fibroblasts, which dampened CXCR4-mediated MDSC recruitment. Using EGFR *v*III-positive xenografts, combination therapy of olaparib and EGFR *v*III-targeted CAR T-cells enhanced anti-tumor immunity by reducing MDSC migration and improving CD8^+^ T-cell infiltration [144].

The use of proinflammatory cytokines to overcome resistance to CAR T-cell therapy has shown promise. IL-12 limits MDSC-mediated immunosuppression through modulation of the myeloid compartment [145]. In preclinical models, IL-12 was shown to change the cell surface expression of co-stimulatory molecules, reduce T-cell suppression by MDSCs, and enhance dendritic cell and macrophage phenotypes. These results suggest that IL-12 could reprogram MDSCs toward antigen-presenting cells [145,146]. Originally, treatment with IL-12 was an attractive method for enhancing anti-tumor immunity as IL-12 can also promote the production of IFN-γ, enhance natural killer (NK), and T-cell cytotoxic functions [147]. However, the NK cell-mediated toxicities associated with systemic IL-12 administration have restricted its use for cancer therapy [148,149,150].

Several studies have been conducted investigating the safety and efficacy of IL-12-secreting CD8^+^ T-cells or CAR T-cells in mice and humans [151,152,153,154]. While some studies observed significant toxicities, others report high safety, suggesting that integrating safety mechanisms to control IL-12 secretion by CAR T-cells may be useful. Additionally, Glassman et al. showed that IL-12 partial agonists could effectively promote anti-tumor immunity while eliminating NK cell-mediated toxicities [155]. Taken together, IL-12 has re-emerged as a potential approach to inhibit the suppressive effects of MDSCs and enhance the anti-tumor immunity of CAR T-cells.

Other CAR-based therapies have emerged showing effectiveness in overcoming various immune escape mechanisms, including CAR-NK cells [156,157]. Moreover, specific CAR-NK cells have demonstrated effectiveness in depleting MDSCs [158]. MDSCs are found to overexpress NKG2D ligands, which are capable of activating the NKG2D cytotoxicity receptor on NK cells [159,160]. The immunosuppressive tumor environment limits this activation; however, this can be overcome by engineering a chimeric receptor that fuses this NKG2D activating receptor to CD3ζ in order to enable NKG2D.ζ CAR-NK cells to eliminate MDSC in vivo [158,160]. Additionally, combining the NKG2D.ζ CAR-NK cells with GD2.CAR-T (specific to the tumor antigen GD2) therapy increased the anti-tumor activity of the CAR-T cells. Unfortunately, engineering these CAR-NK cells has proven difficult as they can be resistant to conventional transgene delivery [161]. With the rapid advancements in CRISPR/Cas-9 technology and precise genetic modifications, this provides a powerful approach to overcome this limitation in CAR-NK cell engineering and has the potential to facilitate other methods of targeting MDSCs [162].

MDSCs and precursor myeloid cells express CCR2, a receptor for CCL2, which is a prevalent chemoattractant in MDSCs recruitment to the tumor [163]. CCR2 blockade can potentially reduce the regulatory myeloid compartment. However, CCR2 also promotes T-cell recruitment, indicating that CCR2 blockade might affect T-cell migration to the tumor [164,165,166]. Nevertheless, CCR2 inhibition appears to affect the regulatory immune cell compartment preferentially and displays promising anti-tumor effects [167]. In many preclinical models, including melanoma, breast, and bladder cancers, CCR2 antagonism coupled with anti-PD-1 therapy enhanced the anti-tumor response of mice compared to anti-PD-1 monotherapy [168]. Tumor-bearing mice treated with combination therapy showed significant increases in CD8^+^ T-cell recruitment and activation, and reduced intratumoral Tregs [168]. While the quantity of MDSCs was not measured in this study, a Phase Ib clinical trial in pancreatic cancer patients has shown that CCR2 inhibition with CCR2-specific antagonist CCX872 decreases monocytic MDSCs (NCT02345408). Moreover, patients receiving CCX872 with FOLFIRINOX chemotherapy show better responses than those receiving chemotherapy alone [169,170].

Blocking MDSC differentiation is another therapeutic strategy currently under investigation. VEGF and STAT3 are important in MDSC differentiation and expansion [171]. Sunitinib is a tyrosine kinase inhibitor that appears to have a potent effect in depleting MDSCs through VEGFR blockade and inhibition of STAT3 activity, among other mechanisms [172]. Renal cell carcinoma patients treated with sunitinib displayed reduced MDSCs and Tregs in the peripheral blood, which inversely correlated with Th1 responses [173]. Moreover, adding sunitinib to patient MDSCs in culture reduced their suppressive capacity and viability, with no effect on MDSCs maturation [173].

Currently, sunitinib is used for the treatment of metastatic renal cell carcinoma and gastrointestinal stromal tumors. It is also being investigated in clinical trials in other metastatic settings [174,175]. The treatment of tumors with sunitinib and immunotherapies is a promising combination therapy in some cancer types. Using a renal cell carcinoma mouse model, treatment with both sunitinib and anti-PD-1 demonstrated enhanced CD8^+^ T-cell infiltration and cytotoxic function and a significant reduction of MDSCs in the TME compared to monotherapies [176]. The combination of nivolumab and sunitinib has demonstrated efficacy in a Phase Ib/II clinical trial for the treatment of patients with advanced soft tissue sarcomas [177]. After six months, the mean progression-free survival rate (mPFSR) and overall survival (mOS) of patients treated with the combination therapy were 48% and 24 months, respectively. This is comparable to another study, Alliance-091401, in which patients with soft tissue sarcomas were treated with nivolumab monotherapy. In this study, the mPFSR and mOS were reported at 15% and 10.7 months, respectively [178]. These results indicate that depleting MDSCs and rescuing anti-tumor activity of effector immune cells could enhance therapeutic response and overcome ICI resistance mediated by MDSCs.

As mentioned previously, MDSC-derived exosomes (M-exo) can contribute to tumorigenesis by carrying cytokines and proteins that promote immunosuppression [33]. Therefore, either the prevention of M-exo formation, release, or binding can potentially mitigate its effects. A brute force method for removing M-exo would be to perform extracorporeal hemofiltration of exosomes from the entire circulatory system. While removing M-exo, this method would remove tumor-derived exosomes as well, which are also detrimental to cancer prognosis [179]. Whether this approach is feasible in a clinical setting is still to be determined. Surface proteins on M-exo that are important in recruitment may be targeted by antibodies. Antibody-mediated blocking of S100A8, S100A9, thrombospondin-1, and its binding partner CD47 were found to decrease the migration of M-exo in vitro [180,181]. More studies need to be done to further explore this novel therapy.

Nanoparticle-based therapies have become a promising method to enhance anti-tumor immunity. Nanoparticles can also be used to decrease the frequency of MDSCs by genetic reprogramming. Tumor-Educated Myeloid Cells (TEMC), which are abundantly found in the TME, can be targeted by nanoparticles harboring short hairpin RNA (shRNA) against STAT3 and C/EBPβ. These nanoparticles can efficiently transfect TEMC and downregulate the expression of STAT3 and C/EBPβ, which are transcription factors essential for MDSC development. In preclinical models of colon carcinoma, the injection of these nanoparticles can increase the effectiveness of anti-tumor vaccines [182]. Nanoparticles such as RNA aptamers can additionally be engineered to deliver chemotherapy agents to targets cells. RNA aptamers against tumor infiltrating myeloid cells conjugated to doxorubicin were found to be more effective in prolonging the survival of fibrosarcoma and breast cancer mouse models compared to free doxorubicin. As MDSCs are most abundant in the TME, these doxorubicin conjugated aptamers can eliminate both MDSCs and tumor cells [183].

Some nanoparticle-based therapies actually target the mechanisms of immunosuppression employed by MDSCs. Using biomimetic engineering, Liu et al. have developed an MDSC membrane-coated magnetic Fe_3_O_4_ nanoparticle (MNP) called MNP@MDSC [184]. By effectively transferring the membrane of a cell such as an MDSC to the surface of a nanoparticle, important features of the cell membrane such as surface antigens, receptors, and enzymes can be exploited to target the TME [184,185]. In preclinical studies, MNP@MDSCs are shown to act as a photothermal therapy agent when the tumor site is irradiated with infrared (IR) lasers. This is attributed to the ability of the MNP to convert IR light into heat, enabling the immunogenic cell death of tumor cells. Additionally, the MNP@MDSCs demonstrate ability to induce state switching similar to macrophages, leading to M1 polarization. These nanoparticles are shown to further enhance anti-tumor immunity by reducing the metabolic activity of the tumor cells [184]. Therefore, while MDSCs contribute to pro-tumor immunity, certain characteristics such as their high affinity for the TME and mechanisms of immune escape can be harnessed for therapy.

Since M-MDSCs and PMN-MDSCs can use different mechanisms to exert their immunosuppressive functions, it is reasonable that therapies could affect these populations differently. In fact, some of the therapies and combination strategies previously discussed preferentially affect one subpopulation. The combination of chemotherapy and VEGF inhibitor bevacizumab as well as anti-Gr1 antibodies were found to deplete only PMN-MDSCs [136,139]. Conversely, the M-MDSC population was reduced with CCR2-specific agonists and engineered CD8+ T-cells secreting IL-12 (NCT02345408) [146]. Many of the other therapies including peptibodies targeting S100 proteins, IL-2/anti-CD40 and GO demonstrated efficacy in targeting all subtypes of MDSCs [139,140,143]. 5FU chemotherapy and sunitinib treatments affected both populations, but M-MDSCs demonstrated greater sensitivity than PMN-MDSCs [137,176]. Rationale for the preferential targeting of specific MDSC populations is limited, demonstrating a need for further basic science research. Understanding these mechanisms could provide rationale for using specific therapies within a clinical setting.

## 6. Conclusions

While MDSCs play a prominent role in the immunosuppression of anti-tumor responses, the mechanisms by which MDSCs influence T-cell activations are becoming more evident. MDSCs can recruit other immunosuppressive cells such as Tregs and can engage inhibitory receptors on T-cells, as well as releasing immunosuppressive ROS and cytokines. These effects can render certain immunotherapies ineffective. However, novel strategies such as cytokine agonists and targeted immunotherapies are being developed to target MDSCs and can be used in combination with existing immunotherapies to improve treatment efficacy.

## Figures and Tables

**Figure 1 cells-10-01170-f001:**
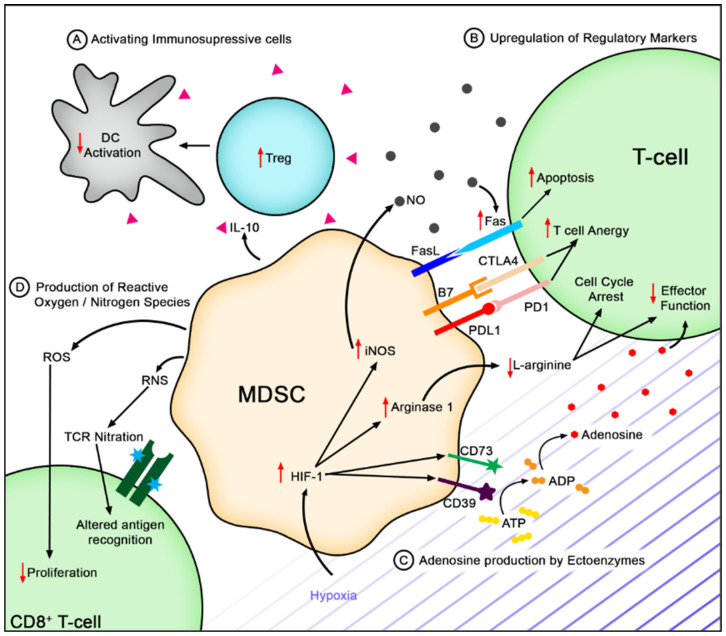
Mechanisms of Immunosuppression in the tumor microenvironment. (**A**) MDSCs activate immunosuppressive cells such as Tregs via IL-10 secretion. These cells can dampen antigen presentation capabilities by dendritic cells, which leads to reduced T-cell activation. (**B**) MSDCs can induce the upregulation of checkpoint molecules such as CTLA4 and PD1 on T-cells, further inducing T-cell anergy, or Fas that can induce T-cell apoptosis. (**C**) Hypoxia in the tumor microenvironment can induce the upregulation of CD73 and CD39 that increases adenosine. (**D**) MDSCs produce ROS and RNS that can decrease T-cell proliferation and alter antigen recognition capabilities. CD8^+^ T-cells and induced the formation of ROS.

**Figure 2 cells-10-01170-f002:**
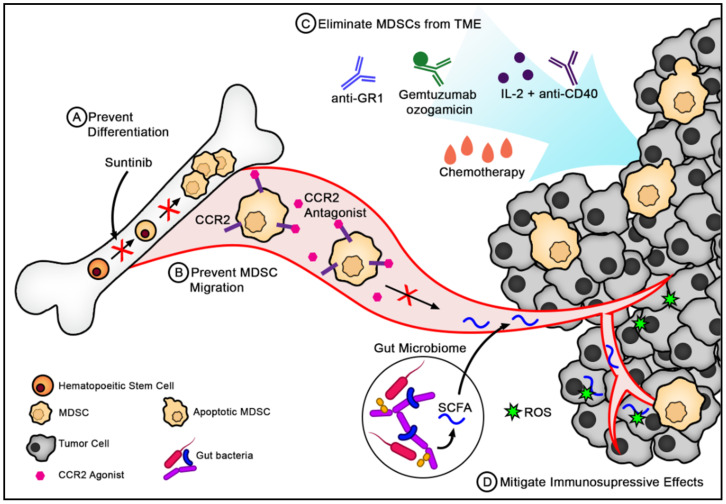
Strategies for targeting MDSCs in cancer. (**A**) Preventing differentiation of hematopoietic stem cells into MDSCs. Sunitinib is a tyrosine kinase inhibitor that inhibits VEGF and STAT3 activity, both of which are crucial in the differentiation and expansion of MDSCs. (**B**) Preventing MDCSs from migrating to the tumor. Chemokines such as CCL2 are critical for the recruitment of MDSCs to the TME. As CCL2 binds CCR2 on the surface of MDSCs, specific CCR2 antagonists can hinder the migration to MDSCs toward the TME. (**C**) Eliminating MDSCs from the tumor by using a variety of immunotherapies and chemotherapy. MDSCs express surface molecules such as Gr1, CD33, and CD40, which provide a target for MDSC depleting antibodies such as anti-Gr1, gemtuzumab ozogamicin (GO), and anti-CD40, respectively. While chemotherapies such as 5-fluorouracil do not specifically target MDSCs, they can effectively eliminate MDSCs through apoptosis. (**D**) Mitigating the immunosuppressive effects of MDSCs at the tumor site. The gut microbiota is capable of producing short-chain fatty acids (SCFAs), which can limit the toxicity associated with ROS produced by MDSCs.

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
