# Peer review of "Immunosuppressive Effects of Myeloid-Derived Suppressor Cells in Cancer and Immunotherapy"

_cells, 2021, doi:10.3390/cells10051170_

Round 1

Reviewer 1 Report

In this review, Mithunah Krishnamoorthy and co-authors first discussed  MDSC biology in terms of phenotype and functions, followed by describing MDSC activities in cancer they focused the impact of these regulatory cells on affecting cancer immunotherapy. These discussions are interesting to the readers of cancer immunotherapy, it covers the relevant literature and raised some exciting discussions, brought some new insights in the therapeutic applications. The manuscript is clearly written and easy to understand. Overall, this is a good review and there are only a couple of issues that require clarification.

  1. Recent advances in the technologies, e.g. single cell RNA sequencing and CytoF, confirmed and revealed the heterogeneity of monocyte/PMNs, including resident macrophages in the circulation or during inflammation/cancer progression. It would be nice that the authors discuss the findings by the advanced technologies in regarding the MDSC diversity and their roles in cancer therapies.
  2. Figure 2 is very comprehensive; however, the figure legend is quite simple. A detailed elucidation on the figure may be helpful for the general reader.
  3. In chapter 1 the authors reported, “M-MDSCs can differentiate into macrophages that are functionally different from traditional macrophages”. It is unclear sentence, The authors should explain better or add a key reference.
  4. In chapter 1, the authors should better describe the key role of c-EPBbeta as main driver of MDSC generation
  5. In chapter 1, the authors should describe the pivotal role of apoptosis-associated molecules to define PMN-or M-MDSCs as identified by Murray’s (J.M. Haverkamp, et al., Myeloid-derived suppressor activity is mediated by monocytic lineages maintained by continuous inhibition of extrinsic and intrinsic death pathways, Immunity , 2014) and Bronte’s (A. Fiore, et al. Induction of immunosuppressive functions and NF-kappaB by FLIP in monocytes. Nat Commun, 2018) laboratory.
  6. In chapter 1, STAT3+ARG1+ CD14+ cells identified in PDAC patients (Trovato R, et al. Immunosuppression by monocytic myeloid-derived suppressor cells in patients with pancreatic ductal carcinoma is orchestrated by STAT3. J Immunotherap Cancer. (2019) 7:255. doi: 10.1186/s40425-019-0734-6) should discuss as potential biomarker also for metastatic disease
  7. In chapter 5, the authors should discuss nanoparticle-based therapies to limit MDSCs by genetic-reprogramming (S. Zilio, et al. 4PD Functionalized Dendrimers: A Flexible Tool for In Vivo Gene Silencing of Tumor-Educated Myeloid Cells, J Immunology 2017) or as chemotherapy cargo (De La Fuente A., Aptamers against mouse and human tumor-infiltrating myeloid cells as reagents for targeted chemotherapy, Scie Transl Med, 2020)

Minor

Line 203. Typo: O2-  needs to replace with O2-

Author Response

Reviewer #1 comments 

In this review, Mithunah Krishnamoorthy and co-authors first discussed MDSC biology in terms of phenotype and functions, followed by describing MDSC activities in cancer they focused the impact of these regulatory cells on affecting cancer immunotherapy. These discussions are interesting to the readers of cancer immunotherapy, it covers the relevant literature and raises some exciting discussions, bringing some new insights in the therapeutic applications. The manuscript is clearly written and easy to understand. Overall, this is a good review and there are only a couple of issues that require clarification.

We thank the reviewer for their positive comments.

1. Recent advances in the technologies, e.g. single cell RNA sequencing and CytoF, confirmed and revealed the heterogeneity of monocyte/PMNs, including resident macrophages in the circulation or during inflammation/cancer progression. It would be nice that the authors discuss the findings by the advanced technologies in regarding the MDSC diversity and their roles in cancer therapies.

Indeed, these advanced techniques have deepened our understanding of MDSC heterogeneity. We have included a brief discussion of the implications of these technologies on lines 48-57. 

2. Figure 2 is very comprehensive; however, the figure legend is quite simple. A detailed elucidation on the figure may be helpful for the general reader.

We thank the reviewer for this suggestion and have provided additional information to the Figure 2 legend to provide more context and clarity. (lines 449-461)

 3. In chapter 1 the authors reported, “M-MDSCs can differentiate into macrophages that are functionally different from traditional macrophages”. It is an unclear sentence, The authors should explain better or add a key reference.

We thank the reviewer for this comment. We have revised the sentence (line 76) for clarity.

 4. In chapter 1, the authors should better describe the key role of c-EPBbeta as main driver of MDSC generation

We thank the reviewer for this suggestion. We have added a brief discussion of C/EPBβ and its importance in MDSC development (line 104-113)

5. In chapter 1, the authors should describe the pivotal role of apoptosis-associated molecules to define PMN-or M-MDSCs as identified by Murray’s (J.M. Haverkamp, et al., Myeloid-derived suppressor activity is mediated by monocytic lineages maintained by continuous inhibition of extrinsic and intrinsic death pathways, Immunity , 2014) and Bronte’s (A. Fiore, et al. Induction of immunosuppressive functions and NF-kappaB by FLIP in monocytes. Nat Commun, 2018) laboratory.

 We agree that this should be an important point of discussion in the manuscript. We have added a brief discussion of the roles of the anti-apoptotic proteins cFLIP and MCL-1 and how they contribute to M-MDSC and PMN-MDSC development respectively (line 114- 121).

6. In chapter 1, STAT3+ARG1+ CD14+ cells identified in PDAC patients (Trovato R, et al. Immunosuppression by monocytic myeloid-derived suppressor cells in patients with pancreatic ductal carcinoma is orchestrated by STAT3. J Immunotherap Cancer. (2019) 7:255. doi: 10.1186/s40425-019-0734-6) should discuss as potential biomarker also for metastatic disease

This finding is very interesting, and we have included it in our discussion of STAT3 in chapter 1 of this manuscript (lines 97-104). We thank the reviewer for this suggestion. 

7. In chapter 5, the authors should discuss nanoparticle-based therapies to limit MDSCs by genetic-reprogramming (S. Zilio, et al. 4PD Functionalized Dendrimers: A Flexible Tool for In Vivo Gene Silencing of Tumor-Educated Myeloid Cells, J Immunology 2017) or as chemotherapy cargo (De La Fuente A., Aptamers against mouse and human tumor-infiltrating myeloid cells as reagents for targeted chemotherapy, Scie Transl Med, 2020)

We thank the reviewer for this suggestion. We have included the articles mentioned to highlight other nanoparticle based therapies to target MDSCs (see lines 585 -596).

Line 203. Typo: O2-  needs to replace with O2-

We thank the reviewer for alerting us to this typographical error. We have now corrected this.

Reviewer 2 Report

The review entitled 'IMMUNOSUPPRESSIVE EFFECTS OF MYELOID-DERIVED SUPPRESSOR CELLS IN CANCER AND IMMUNOTHERAPY' describes the roles of MDSC's and their potential role in Immunotherapy.

The review article is very interesting, well written, and describes the multifaceted roles of MDSCs with latest advancements in the field. The review is very intriguing, especially, the section related to Combination strategies targeting MDSC’s. The authors could discuss the below advancements in the field.

Considering the landscape of tumor microenvironment and the role of MDSC's in the regulation of Tregs and other cytokines, MDSC's are an attractive therapeutic target.

1.  As MDSC’s promote tumorigenesis, the authors had described blocking the effects of VEGF and STAT3, It would be interesting to know the role of MDSC’s in EMT.

2. There is evidence suggesting the usage of TGFb with ICIs to counter resistance. The authors can discuss it. Below are some references                                                               

A. TGF-β attenuates tumour response to PD-L1 blockade by contributing to exclusion of T cells. Nature. 2018.                                                                                                                     

 B. An experimental model of anti-PD-1 resistance exhibits activation of TGFß and Notch pathways and is sensitive to local mRNA  Immunotherapy,   OncoImmunology,2021

  C. Selective inhibition of TGFβ1 activation overcomes primary resistance to checkpoint blockade therapy by altering tumor immune landscape, Science Translational Medicine  25 Mar 2020:

3. With the recent advancement of CRISPR-Cas9, the authors could suggest alternate approaches to target MDSCs.                                                                                             

The effects of a CRISPR/Cas9 IL-6 knockout in 4T1 mammary carcinoma cells on myeloid-derived suppressor cells (MDSCs) and Th17/Th22 cells, J Immunology

4. Targeting exosomes or nanovesicles released by MDSCs is also an attractive therapeutic mode to inhibit transport of pro-tumorigenic proteins.

5. The authors have nicely described the current advancements, it would be interesting to give an overview of the emerging targets related to G-MDSC’s or M-MDSC's that could be attractive therapeutic targets.

Minor comment

  Line 108 on Page 3 is hidden under figure 1.  

Author Response

Reviewer #2 Comments and Responses

The review entitled 'IMMUNOSUPPRESSIVE EFFECTS OF MYELOID-DERIVED SUPPRESSOR CELLS IN CANCER AND IMMUNOTHERAPY' describes the roles of MDSC's and their potential role in Immunotherapy.

The review article is very interesting, well written, and describes the multifaceted roles of MDSCs with latest advancements in the field. The review is very intriguing, especially, the section related to Combination strategies targeting MDSC’s. The authors could discuss the below advancements in the field.

Considering the landscape of tumor microenvironment and the role of MDSC's in the regulation of Tregs and other cytokines, MDSC's are an attractive therapeutic target.

 We thank the reviewer for their positive comments.

 1. As MDSC’s promote tumorigenesis, the authors had described blocking the effects of VEGF and STAT3, It would be interesting to know the role of MDSC’s in epithelial-mesenchymal transition EMT.

We thank the reviewer for this suggestion. We have added a brief discussion of the role of MDSCs in the EMT (see lines 236-247)

2. There is evidence suggesting the usage of TGFb with ICIs to counter resistance. The authors can discuss it. Below are some references                          A. TGF-β attenuates tumour response to PD-L1 blockade by contributing to exclusion of T cells. Nature. 2018.                                                                            B. An experimental model of anti-PD-1 resistance exhibits activation of TGFß and Notch pathways and is sensitive to local mRNA  Immunotherapy,   OncoImmunology,2021

C. Selective inhibition of TGFβ1 activation overcomes primary resistance to checkpoint blockade therapy by altering tumor immune landscape, Science Translational Medicine  25 Mar 2020:

 We agree that this is an important strategy to discuss. We have added a brief discussion of the role of MDSCs in TGF-β-mediated ICI resistance and the use of TGF-β inhibitors to overcome resistance (See lines 311-332)

3. With the recent advancement of CRISPR-Cas9, the authors could suggest alternate approaches to target MDSCs.                                                                    The effects of a CRISPR/Cas9 IL-6 knockout in 4T1 mammary carcinoma cells on myeloid-derived suppressor cells (MDSCs) and Th17/Th22 cells, J Immunology

We thank the reviewer for this suggestion. However, the indicated article does not directly show the use of CRISPR-Cas9 technology in  directly targeting MDSCs. CRISPR-Cas9 is currently being used in engineering CAR based therapies, and is found to overcome some limitations of CAR-NK cell generation. Since CAR-NK cells have previously been engineered to target MDSCs, we have included a brief discussion of CAR-NK mediated MDSC depletion and some benefits of CRISPR-Cas9 technology. (See lines 519 - 532)

4. Targeting exosomes or nanovesicles released by MDSCs is also an attractive therapeutic mode to inhibit transport of pro-tumorigenic proteins.

Indeed, MDSCs release nanovesicles that can lead to immunosuppression. Although therapies specifically targeting MDSC derived nanovesicles have yet to be established in humans, we have discussed some potential strategies.(see line 145-146, 156 and, 572-582 ).

5. The authors have nicely described the current advancements, it would be interesting to give an overview of the emerging targets related to G-MDSC’s or M-MDSC's that could be attractive therapeutic targets.

We thank the reviewer for this suggestion. Please refer to lines 248-262 in which the different mechanisms used by M-MDSCs and PMN-MDSCs are discussed. Additionally, we have added a brief discussion of how the currently therapies and combination strategies affect each subpopulation (See lines 611-624)

Minor comment

 Line 108 on Page 3 is hidden under figure 1.  

We thank the reviewer for alerting us to this error. We have now corrected it.